# Efficacy of Articaine or Eugenol for Pain Relief after Emergency Coronal Pulpotomy in Teeth with Irreversible Pulpitis: A Randomized Clinical Trial

**DOI:** 10.3390/dj11070167

**Published:** 2023-07-11

**Authors:** Fernandez De Grado Gabriel, Bourdin Clément, Fioretti Florence, Musset Anne-Marie, Offner Damien

**Affiliations:** 1I INSERM (French National Institute of Health and Medical Research), UMR 1260, Regenerative Nanomedicine (RNM), FMTS, CRBS, 1 Rue Emile Boeckel, 67084 Strasbourg, France; 2Université de Strasbourg, Faculté de Chirurgie Dentaire, 8 rue Ste Elisabeth, 67000 Strasbourg, France; 3Hôpitaux Universitaires de Strasbourg (HUS), Pôle de Médecine et Chirurgie Bucco-Dentaires, 1 place de l’hôpital, 67000 Strasbourg, France

**Keywords:** pain, pulpitis, analgesia, articaine, eugenol, emergency treatment

## Abstract

Irreversible pulpitis is an extremely painful dental pathology. Its emergency treatment, pulpotomy, should include the use of a pulp dressing in the pulp chamber until the final treatment. Various antalgic products have been suggested as efficient medications to relieve the patient’s pain and are commonly used, but data for scientific validation are scarce. Objective: We evaluated the efficacy of articaine or eugenol in the diminution of pain after pulpotomy. Design: We included 100 patients with irreversible pulpitis and evaluated their initial pain levels. Pain was measured using a 0–10 numeric rating scale. After treatment through pulpotomy, we randomized them into two groups using either articaine or eugenol as a pulp dressing and evaluated their pain level at 1, 3 and 7 days. Results: Initial pain levels were severe (7.53). The use of painkillers was not associated with lower levels of pain. Both treatments showed great efficiency on day 1, with stronger efficiency of eugenol than articaine, showing a decrease of 6.24 versus 4.89 (*p* = 0.025). Both treatments were efficient, whatever the age or sex of the patient, the initial pain level, and the causal tooth. Conclusion: Pulpotomy is an efficient way to relieve pain, using either articaine or eugenol. When choosing a pulp dressing, eugenol should be the first choice.

## 1. Introduction

Irreversible pulpitis is a frequent oral pathology that leads to dental pulp destruction. It accounts for up to 35% of painful dental emergencies, with an elevated prevalence among young adults [1,2,3].Around 50% of adult patients suffering from irreversible pulpitis are under 35 years old [1,2,3,4]. While not life-threatening, it is an extremely painful process (around 7, 5/10 on the visual analog scale or numeric rating scale) and an extremely frequent cause for emergency appointments with a dental surgeon [3,5,6,7]. Despite growing evidence that a well-performed pulpotomy using a biocompatible material such as Mineral Trioxide Aggregate (MTA) or Biodentine could achieve a comparable success rate [8], the complete and current recommended treatment is a full endodontic (root canal) treatment, which includes a full pulpectomy and root filling and takes up to a few hours [7,9,10]. While it is theoretically possible to perform it immediately, it is often impracticable during a short emergency appointment that may last between 15 and 30 min [11,12]. Another emergency treatment, sufficient to relieve the pain, is the removal of the whole pulp in the pulp chamber, named pulpotomy, with the application of a sedative pulp dressing covering the root pulp and a temporary filling [10,11,12,13]. Pulpotomy is a fast intervention of less than half an hour, which is extremely efficient and allows for a delay in performing the needed full endodontic treatment during an ulterior appointment [10,11,12,13,14].

Various products have been used as temporary pulp dressings over the years [10,13]. Associations of medications have been used for a long time, but most of them are controversial and no longer available in many countries, mostly due to a suspicion of carcinogenic effects or toxicity from some of their components, such as formocresol [15].

The evaluation of various medications used as pulp dressings has shown no difference in the efficiency of the evaluated products (zinc oxide–eugenol cement, a cotton pellet either dry or moistened with camphorated phenol, cresatin, eugenol, or isotonic saline), all of them being highly efficient in relieving pain [13]. However, the only study comparing different medications is ancient, had low statistical power, with only up to 12 patients for each treatment group and low precision in pain measurement (“pain”, “discomfort”, or “no symptom”), and was thus unlikely to detect differences in treatments efficiencies [13].

With the disappearance of some medications, we wanted to validate and compare the efficiency of two inexpensive, easy-to-use and easily available products used for analgesia, which are present in every dental office: articaine and eugenol. Pulpotomy by itself seems to be an efficient treatment even without using any active pulp dressing [11,13]. However, we could not legally evaluate the efficiency of a dry or wet cotton pellet as a control, in accordance with the recommendations of the French health agency (HAS), which states that a pulp dressing must be used [9].

Articaine is an anesthetic drug commonly used for local and loco-regional anesthesia and, as such, available during the procedure. Its efficiency as an anesthetic has been extensively studied and seems slightly superior to other dental anesthetic drugs in some situations, like inferior alveolar nerve block [16,17], but we did not find any studies evaluating its efficiency as pulp dressing. Articaine can be expected to achieve anesthesia as a pulp dressing. Despite a relatively short half-life (around 1 h), its progressive release from a cotton pellet could increase the anesthesia duration [18].

Eugenol (4-Allyl-1-hydroxy-2-methoxybenzene) has complex dose-dependent effects on inflammation, apoptosis and anesthesia and is widely used for its antibacterial, antiviral, antioxidant, anti-inflammatory and analgesic properties, per se or as part of more complex materials such as zinc-oxide–eugenol cements [19]. Its use as an antalgic dressing on inflammatory dental pulp is well documented. While theoretically potentially cytotoxic and allergenic with rare reports of allergic reactions, this is not a concern in the studied situation in which a low volume of eugenol is kept enclosed in the teeth, in contact with the radicular pulp whose removal should happen a few days or weeks later [13,19,20,21,22]. A study even confirmed that using a eugenol-based dressing may be a reliable option for long-term interim management of irreversible pulpitis for around 6 months while waiting for an endodontic treatment [21].

The aim of this study is to evaluate the efficacy of articaine and eugenol as pulp dressings for pain control by comparing pain diminution after a pulpotomy following a diagnosis of irreversible pulpitis in a double-blind (blind patient and phone investigator) randomized clinical trial.

## 2. Materials and Methods

The study protocol, written with respect of the declaration of Helsinki, was validated by an ethics committee according to French law (Patient Protection Comity CPP): “Comité de protection des personnes Est IV”; N° IDRCB: 2017-A03416-47/RIPH3 2017-HUS N°6953, on 14 March 2018. The study was also approved by the institutional review board of Strasbourg’s Hospital: The scientific council of the Direction of Clinical Research and Innovation (DRCI) and registered as “HUS N°6953”. The trial was registered with the clincialtrials.gov identifier NCT03472456 (21 March 2018). The National Commission for data processing and Freedom (CNIL) and the National Health and Drug Association (ANSM) were, respectively, informed of the data collection and of the study protocol. We followed the CONSORT Guidelines (available as Appendix A). Prior to any participation, patients were informed and signed a written consent explaining the whole protocol. 

The study took place in the emergency dental care unit (CASU) of the Strasbourg (France) University Hospital (HUS).

The null hypothesis was “articaine and eugenol have a similar efficacy on pain diminution after treatment on day 1, 3 and 7”.

Our main judgment criterion was the evolution of pain measured using the 0 to 10 numeric rating scale for pain (NRS) score 24 h after treatment. We also measured NRS scores 3 and 7 days after treatment.

Our inclusion criteria were: informed and willing adult (>18 years old) patient with social insurance, a sufficient understanding of the French language to answer our questions with no risk of error, not undergoing antalgic treatment for another pathology, diagnosed with an irreversible pulpitis (see below) on any mature permanent tooth and for which pulpotomy was indicated and achievable. We did not include patients on whom we were not able to perform the pulpotomy (because of health issues, for example, risk of infectious endocarditis), patients whose pain was caused by two or more teeth and patients undergoing chronic antalgic medical treatment.

Diagnosis of irreversible pulpitis was performed by a senior dental surgeon on the following criteria: Spontaneous, severe and prolonged pain, exaggerated following cold exposure on a single tooth, presenting a logical etiology (such as carious disease) and no sign of infection nor pulp necrosis. Radiographs were used to confirm the presence of a logical etiology and eliminate teeth with signs of necrosis, such as visible periapical radiolucent lesions. We explained the protocol and anonymity to eligible patients. Signed consent was obtained before we started the treatment procedure. Due to the risk of procedure or anesthetic failure, patients were considered, as included only after the success of the pulpotomy procedure, just before the randomization. 

On day 0 before the treatment, included patients answered a questionnaire in which we recorded the spontaneous pain level using NRS and the phone number of the patient. Other evaluated variables were age, sex, treated tooth, consumption of painkillers in the past 24 h, medical conditions and treatments. 

Patients were then treated for irreversible pulpitis using the standard pulpotomy procedure performed by two dental students in their 5th of 6th year of studies who undertook theoretical and practical courses about endodontics and pulpotomy, supervised by a senior investigator (D.O. or G.FdG.) using the following steps: -Anesthesia using articaine from a 40 mg/mL anesthetic cartridge with 1/200,000 vasoconstrictor (adrenaline). The Inferior Alveolar Nerve Block technique was used in the first intention for mandibular molars, and vestibular infiltration for other teeth. If not enough, intra-ligamentary anesthesia was performed, or intra pulpal if the pulp was exposed.-The tooth was isolated from the saliva using a rubber dam when the use of a clamp was possible. If using a rubber dam was impossible (e.g., structurally compromised tooth) and since pre-endodontic restoration was not manageable in an emergency, and isolation was obtained using cotton rolls and dental aspirator.-Access to the pulp chamber was achieved using a diamond bur; destruction of the whole pulp in the pulp chamber using an Endo-Z bur. Sodium hypochlorite on a cotton pellet was used to clean the chamber for a few seconds. Complete pulpotomy was confirmed by the end of any bleeding and the visualization of the root canal entrances. In case of persistent bleeding, compression using a dry cotton pellet was performed for 2 min.

After a senior confirmed the success of the pulpotomy procedure, patients were included in the study and were randomly and blindly (blind patient, the practitioners could not be blinded) assigned a medication: either a drop of articaine from a 40 mg/mL anesthetic cartridge with 1/200,000 vasoconstrictor (adrenaline) commonly used for anesthesia (Septanest; Septodont, Saint-Maur-des Fossés, France) or a drop of pure eugenol (Eugenol; Produits dentaires SA, Vevey, Switzerland) on a cotton pellet covered by a temporary filling material without eugenol (Cavit; 3M, Cergy-Pontoise, France). 

Failure to perform the pulpotomy procedure did not lead to inclusion.

No painkiller or any other drug was prescribed, but patients were informed that they were allowed to use painkillers if needed. 

A blind investigator called them back by phone on day 1, Day 3 and Day 7 to evaluate pain and the use of painkillers. 

At day 0, patients were offered an appointment to perform the full endodontic treatment if it was deemed possible or an extraction if it was not after a one-month delay after the last phone call.

We evaluated the necessary number of patients to 100 to assess a difference in pain relief of 2 on the NRS between the two groups, with less than 10% of patients lost to follow-up, using the standard risks α = 0.05 and β = 0.10 and estimating the standard deviation for pain to be of 3 at most based on a previous study [7]. 

The flow diagram of this study is shown in Figure 1.

Statistical analysis was made using the software R [23] (R; R Core Team, Vienna, Austria). Block randomization was performed for 10 blocks of 10 patients using a random function in a Microsoft Excel document. A sheet with 100 hidden cells was generated. For each inclusion, the current investigator asked the statistician to reveal a new cell that determined the group allocation. Comparisons of mean pain levels between the two groups were made using Wilcoxon–Mann–Whitney’s test since the distribution of the “pain level” variable was not normal. The same test was used to compare the mean age of the sample between the two groups at baseline. Analysis of variance was used for the comparison of means between more than two groups. Comparisons of two quantitative variables (such as age and pain or pain on day 0 and pain on day 1) were made using Pearson’s correlation test.

## 3. Results

We included 100 patients (Table 1). We lost nine patients to follow-up over the first day and 17 over the 7 days period. The characteristics of those patients were similar to the rest of the sample, especially considering the initial pain level. We excluded two patients who had an initial NRS pain score of 3, which was deemed too low for a real diagnosis of irreversible pulpitis. This exclusion criterion had not been anticipated in the protocol. We also excluded a patient whose diagnosis was deemed erroneous after a second appointment during the protocol, which resulted in a diagnosis of apical periodontitis and not irreversible pulpitis.

Two more patients were excluded from the day 7 analysis because they had another emergency appointment due to persistent pain during the time of the study. Those two patients were part of the articaine group and showed a significantly smaller diminution of pain than the other participants on days 1 and 3. We simulated the best (NRS score at 0 on day 7) and the worst (no diminution of pain) situations for the missing data of those two patients, which made no statistical difference. We thus excluded them from the final analysis concerning the evolution of pain at the 7th day only (Figure 1).

No adverse effects were reported during or after the study outside those three exclusions: one diagnostic failure and two for where the treatment was inefficient and likely failed.

The sample used for baseline and day 1 analysis is composed of 86 patients (Table 1), while the analysis for day 7 is performed on the 78 remaining patients.

Our 86-patient sample was composed of 46 women (53%) and ranged from 18 to 71 years old (mean 31). Among them, 33 (37%) took painkillers before inclusion. 

Causal teeth were mostly mandibular molars and maxillary molars, followed by other maxillary teeth and other mandibular teeth. Initial pain levels were not significantly different between groups (Table 1).

Among those 86 patients, the mean initial pain NRS score was 7.53 (±1.57), with a small non-significant (*p* = 0.157) difference between the two groups: 7.29 CI95% [6.73–7.95] for the eugenol group and 7.77 CI95% [7.37–8.17] for the articaine group (Figure 2). 

Neither sex, nor age, type and location of tooth nor painkiller use had any significant association with initial pain level nor were they different between the two groups (Table 1). 

On day 1, pain among the whole group significantly dropped from 7.53 (±1.57) to 1.99 (±2.47) (*p* < 0.001). Most patients (91.9%) experienced a decrease in pain, 75.6% of the patients had a pain score under 3, and 45.3% had no more pain at all. Pain intensity on day 1 was not correlated with pain intensity on day 0 (*p* = 0.43).

Pain decreased by 6.24 (±2.68) to 1.05 CI95% [0.51–1.58] among the eugenol group and by 4.89 (±2.81) to 2.89 CI95% [2.05–3.72] among the articaine group, showing a significant decrease in both groups (*p* < 0.001 in both groups) as well as a significantly higher decrease (*p* = 0.025) and a significantly lower pain perceived on day 1 (*p* < 0.001) among the eugenol group (Figure 2).

On days 3 and 7, the pain further decreased among both groups (Figure 2). Final measures on day 7 among 78 patients showed that 70.5% of them had no pain at all, and 89.7% had a pain score of 3 or under. 

The consumption of painkillers on days 1 (24% of the population), 3 (20%) and 7 (13%) was significantly associated with a higher level of pain: 4.05 vs. 1.32 on day 1 (*p* < 0.001), 4.47 vs. 0.71 on day 3 (*p* < 0.001) and 4.60 vs. 0.43 on day 7 (*p* < 0.001), in both articaine and eugenol groups.

Neither age (*p* = 0.22) nor sex (*p* = 0.63) nor the type of causal tooth (*p* = 0.29) was associated with the changes in pain levels between day 0 and day 1. Similar results were obtained on days 3 and 7.

## 4. Discussion

Eugenol causes a stronger decrease in pain levels than articaine on day 1. Pain decreases on days 3 and 7 are similar in both groups, with a decrease of around 5 points on the NRS. The eugenol group keeps lower pain levels than the articaine group on days 3 and 7 due to the stronger decrease on day 1. This is the first study quantifying pain evolution after pulpotomy; previous studies only used qualitative measures such as “Mild pain” [13,14]. 

Both treatments led to a decrease in pain levels for more than 90% of patients and thus confirmed that pulpotomy and the use of a sedative pulp dressing is an efficient emergency treatment for irreversible pulpitis, with more than 45% of the patients experiencing total relief immediately or in less than 24 h. This efficiency was confirmed independently of the tooth location or type and is as good as what was obtained with full pulpectomy and occlusal reduction on molars in another study [7]. 

Using eugenol as a pulp dressing showed a significantly higher reduction in pain level and should be considered as the first choice when performing an emergency pulpotomy if a full pulpectomy cannot be performed, considering that it is cheap and easily available. Articaine is still a valid alternative, considering that it is also easily available since it was likely used as a local anesthesia to perform the pulpotomy. It is likely that other anesthetic products or even pulpotomy without pulp dressing could also achieve good results, but further studies are needed to assess their respective effects. Since no control was used to comply with the recommended procedure, we cannot separate the efficiency of the procedure from the efficiency of the treatments. It is very likely that pulpotomy is already highly efficient for relieving pain, even without using any pulp dressing [11,13].

Initial pain levels were considered as severe according to the numeric rating scale for pain, which is consistent with a diagnosis of irreversible pulpitis, one of the most painful dental pathologies. It is also consistent with pain levels reported in the literature [7]. We observed a mean age of 31 years old, which is coherent with previous studies; irreversible pulpitis is more frequent among young adults [1,4].

Neither tooth type nor location were associated with initial pain level nor pain reduction. Other authors reported a lower efficiency of pulpotomy on molars [14]. Since our goal was to compare two pulp dressings, patients on whom we were not able to perform a pulpotomy due to extreme pain levels not alleviated by anesthesia were not included. Pulpitis in those patients was often localized on mandibular molars, which may explain this difference. 

The use of painkillers before treatment was not associated with the initial pain level. Paracetamol and ibuprofen were the most used painkillers. Only four patients used stronger painkillers: Codeine, Tramadol, or a combination of Paracetamol (88%), Opium (3%) and Caffeine (9%). Those four patients still evaluated their pain at 8 on the NRS. This confirms the low efficiency of painkillers on pain associated with irreversible pulpitis and the need for an emergency treatment such as pulpotomy [24]. It should be noted that despite the low efficiency of those drugs in pain management, they may facilitate the anesthesia during the pulpotomy procedure and are still useful [17].

Use of painkillers after treatment was associated with higher pain levels, which is quite logical since patients were advised to stop taking painkillers once free of pain. This once again assesses the low efficiency of painkillers, even in post-treatment residual pain [10,24].

The main strengths of this study are (1) the large sample when compared to similar studies, (2) the strict, detailed protocol including a reproducible method for pulpotomy and a seven-day follow-up, and (3) the use of the NRS for pain recording, allowing for a more precise description of pain-levels evolution than the qualitative scales found in other studies [13,14].

The main limitation of this study is the lack of a placebo control. Our results are thus limited to a comparison of pulpotomy + eugenol vs. pulpotomy + articaine, allowing us to conclude on the efficiency of eugenol vs. articaine but not on the specific effect of each product independently of the pulpotomy. As stated in the introduction, we had to comply with national recommendations and were not able to use an inactive product as a control. We chose to evaluate two products that are cheap, widely available, and traditionally used for short-term pain management. We did not evaluate the most recent but experimental procedures, such as long-term pulpotomy, using MTA or Biodentine whose goal is mostly long-term survival, which was not the purpose of this study. However, the higher efficiency of eugenol compared to articaine and the overall excellent efficiency of the procedures are still important and valid results. Other studies could evaluate the efficiency of different dressings, such as zinc oxide–eugenol cements, which could deliver eugenol while ensuring the temporary filling of the tooth, or even the absence of any dressing since pulpotomy seems to be efficient in itself due to the removal of the coronal pulp and independently of any pulp dressing [13]. As an emergency treatment, pulpotomy should be followed as soon as possible by a full pulpectomy and endodontic obturation [9].

We also experienced a major attrition of our sample over only seven days, close to the anticipated 10% on day 1 but much higher (32%) on day 7, despite several tries to reach the patients back by phone. This may be due to the specific population attending the emergency dental care unit in Strasbourg, which consists mainly of needy patients with social difficulties and low motivation once the emergency is treated. The missing patients did not display any specific characteristics compared to the rest of the sample and were likely missing at random, not leading to any bias outside a loss of statistical power.

The interventions were performed by supervised dental students in their last or second-to-last year of study. Despite supervision by a senior investigator, this may have introduced variability in the results while also confirming that pulpotomy can be performed with high efficiency and good reproducibility, even by young practitioners.

Only a few recent studies about pulpotomy as a treatment for irreversible pulpitis were found, and even fewer detailing the possible procedures [11,14,21]. This may be due to two factors: (1) the fact that pulpotomy has been used since a long time ago (pulp removal was first mentioned in 1746 by Pierre Fauchard and pulpotomy in 1756 by Philipp Pfaff) with great efficiency [10] and practitioners did not feel the need to evaluate its details again. (2) since pulpotomy is an emergency treatment, it may be disregarded by researchers, and most studies focus on the long-term prognostic of the teeth and on the final treatment: pulpectomy and endodontic obturation. However, emergency pain management is of crucial importance in dentistry since pain impacts the quality of life and is the primary concern of the patient. Each and every factor in reducing the patient’s pain is important.

## 5. Conclusions

Irreversible pulpitis is a painful dental pathology. We confirmed that pulpotomy is an efficient alternative emergency treatment of irreversible pulpitis to relieve the patient’s pain when a full endodontic treatment cannot be immediately performed. While pulpotomy using either articaine or eugenol is always efficient, using eugenol as a pulp dressing seems to be slightly more efficient than using articaine to ensure lower pain levels on days 1, 3 and 7 after treatment and should be preferred if available.

## Figures and Tables

**Figure 1 dentistry-11-00167-f001:**
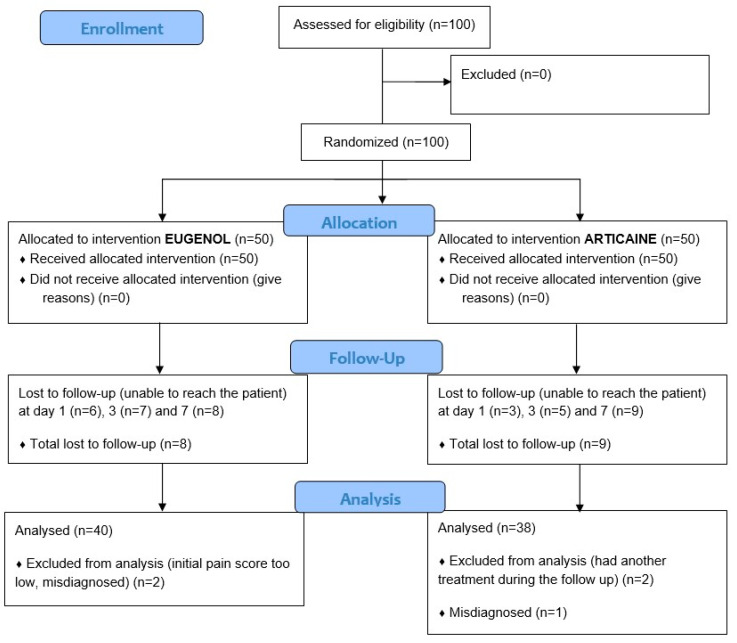
Flow Diagram.

**Figure 2 dentistry-11-00167-f002:**
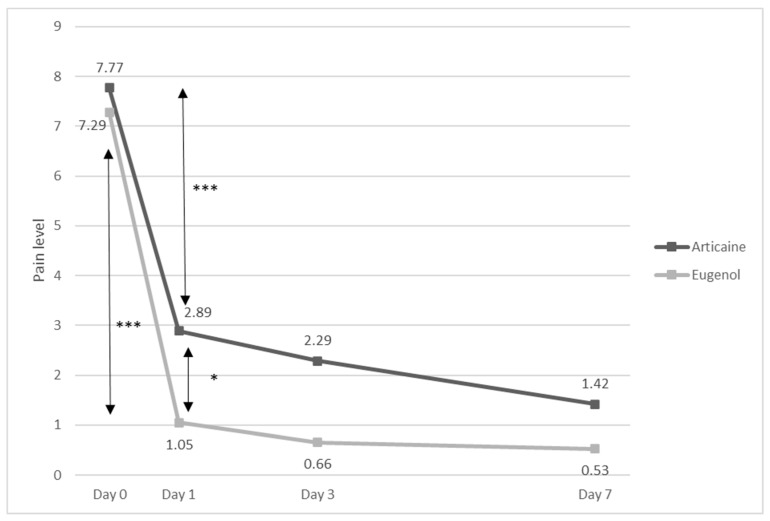
Pain level evolution according to treatment. Decrease between day 0 and day 1 is significant among both groups (*** *p* < 0.001), and there is a significantly higher decrease among the eugenol group (6.24 vs. 4.89, * *p* < 0.05).

**Table 1 dentistry-11-00167-t001:** Sample characteristics at day 0 and mean pain levels. (SD: Standard Deviation; *: *p*-value for correlation between age and mean pain level).

	Overall		Articaine		Eugenol		Articaine vs. Eugenol Pain Level
	n (%)/ mean ± SD	Pain level ± SD	n (%)/ mean ± SD	Pain level ± SD	n (%)/ mean ± SD	Pain level ± SD	*p*
Full sample	86 (100%)	7.53 ± 1.6	44 (51%)	7.77 ± 1.3	42 (49%)	7.29 ± 1.8	0.23
Age (mean)	31.26 ± 10.8	/	31.0	/	31.6	/	0.34 *
Sex							
Women	46 (53%)	7.80 ± 1.3	26 (59%)	7.81 ± 1.5	20 (48%)	7.80 ± 1.5	0.69
Men	40 (47%)	7.22 ±1.8	18 (41%)	7.72 ±1.1	22 (52%)	6.82 ± 2.2	0.10
Tooth location							
Maxillary Molar	24 (28%)	7.17 ± 1.7	11 (25%)	7.63 ± 0.8	13 (31%)	6.77 ± 2.1	0.20
Mandibular Molar	33 (38%)	7.52 ± 1.4	20 (45%)	7.55 ± 1.4	13 (31%)	7.46 ± 1.5	0.89
Other Maxillary Teeth	21 (24%)	7.62 ± 1.7	8 (18%)	8.25 ± 1.4	13 (31%)	7.23 ± 1.8	0.22
Other Mandibular Teeth	8 (9%)	8.5 ± 1.6	5 (11%)	8.2 ± 1.9	3 (7%)	9 ± 1.0	0.76
Painkiller intake before treatment							
Yes	33 (38%)	7.79 ± 1.5	15 (34%)	8.27 ± 1.0	18 (43%)	7.39 ± 1.7	0.17
No	53 (62%)	7.38 ± 1.6	29 (66%)	7.52 ± 1.4	24 (57%)	7.21 ± 1.9	0.55

## Data Availability

The datasets and statistical analysis used and/or analyzed during the current study are available from the corresponding author on reasonable request.

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
