# Peer review of "Efficacy of Articaine or Eugenol for Pain Relief after Emergency Coronal Pulpotomy in Teeth with Irreversible Pulpitis: A Randomized Clinical Trial"

_dentistry, 2023, doi:10.3390/dj11070167_

Round 1
Reviewer 1 Report
Why authors chose these groups, once the literature shows that eugenol has the same padron in the pulp tissue as formocresol, so using these dressing material the tissue will have its nerve endings "mortified".
Also authors could better explain the needs of a dressing before pulpotomy intead use MTA for exemple.
My advice for the authors is to search the literature to impruve the manuscript and study design.
Author Response
Thank you for having taken the time to review our article.
You will find below our answers to your remarks and the changes we made to the manuscript.
Point1: Why authors chose these groups, once the literature shows that eugenol has the same padron in the pulp tissue as formocresol, so using these dressing material the tissue will have its nerve endings "mortified".
Response1: As stated in introduction, pulpotomy was only intended as a temporary treatment and its sole objective was pain management, not conserving the pulpal vitality since it was followed by a full endodontic treatment. Eugenol toxicity is highly dependent on its concentration. While eugenol may not guarantee the conservation of the nerve vitality on the long run, this is not a problem as long as the patient is relieved from pain until his next appointment. A superficial necrosis of the root pulp was deemed acceptable if it ensured pain relief.
Quoting "Markowitz K, Moynihan M, Liu M, Kim S. Biologic properties of eugenol and zinc oxide-eugenol. A clinically oriented review. Oral surgery, oral medicine, and oral pathology. 1992": "At low concentrations eugenol inhibited nerve activity in the reversible manner like a local anesthetic. After exposure to high concentrations of eugenol, nerve conduction was irreversibly blocked, indicating a neurotoxic effect. "
Point 2: Also authors could better explain the needs of a dressing before pulpotomy intead use MTA for exemple.
Response 2: You are right and we tried to mention it in introduction, it seems however that we were not clear enough. We detailed the following sentence to make it clearer that a permanent pulpotomy using MTA could be an alternative to a temporary pulpotomy followed by a full pulpectomy.
"Despite growing evidence that a well performed pulpotomy using a biocompatible material such as Mineral Trioxide Aggregate or Biodentine could achieve comparable success rate (8), the complete and current recommended treatment is a full endodontic (root canal) treatment ..."
Point 3 : My advice for the authors is to search the literature to impruve the manuscript and study design.
Response 3: We made a lot of changes, especially in the results part, following another reviewer's comments and we hope this new version will meet your expectations. However, please note that this is an article about a clinical trial that is already finished and there is no way we could change the design now. We tried our best to highlight the values of this work as well as its limitations.
Thank you again for your reviewing.
Reviewer 2 Report
“Articaine or eugenol for pain relief after emergency coronal pulpotomy in teeth with irreversible pulpitis: a randomized clinical trial” was submitted to Dentistry Journal
The study aimed to evaluate the efficiency of pulpotomy using articaine or eugenol in the diminution of pain.
The authors concluded that when choosing a pulp dressing, eugenol should be the first choice.
The manuscript deals with an interesting issue; however, there are several concerns related to the study.
Title: This RCT assesses efficacy; therefore, it is suggested to indicate it in the title: Evaluation of the efficacy of articaine or eugenol..
Abstract
Objective: It must be reviewed. This RCT does not assess the efficacy of pulpotomy.
Design: The same scale was also used to assess pain on days 3 and 7.
Line 23. The use of painkillers was not associated with lower levels of pain. The analysis does not show clearly how this association was evaluated. Table 1 presents the ingestion of analgesics in each group, but the crossing of variables (pain and analgesics) is not observed. A multivariate analysis is desirable.
Line 25. The pain diminished again on days 3 and 7 in both 25 groups. Please present p-values.
Lines 26-27. Both treatments were efficient whatever the age or sex of the patient, the initial pain level, and the causal tooth. The analysis presented does not allow for these conclusions. Multivariate analysis allows it.
Conclusion: Authors must focus on the results to draw conclusions.
Keywords: “IRREVERSIBLE pulpitis” is not a MeSH term.
Introduction
Lines 32-35. The information presented is based on outdated references.
Line 37. Please review the form to number the references that are consecutive. Keep this in mind throughout the manuscript.
Lines 41-42. Please mention when it is “often impracticable”.
Lines 48-51. "Associations of medications" is indicated; however, only formocresol is mentioned.
Lines 68-69. "Specific situations" should be mentioned.
Lines 66-70. Mention should be made of the scientific argument that a cotton pellet impregnated with articaine can have an analgesic effect on a pulpal remnant.
Line 82. Please consider the recommendations about the objective presented above.
Lines 84-85. It must be removed. It is part of the methodology. Also, consider the term efficacy in the null hypothesis.
Methods
Line 98. Please indicate whether the recommendations made by the Declaration of Helsinki were considered.
Line 126. Indicate how many students participated. Did they have any previous training? If so, detail it.
Line 146. Use of analgesics. To adequately discriminate this variable, a regression analysis is required to adjust for this confounding variable.
Line 147. Blind examinator. Indicate the level of academic training of this examiner.
What relationship does this examinator have with the research group? Please present the calibration process of this operator, the statistical test used, and its results.
Lines 151-154. Was a one-tailed or two-tailed test used?
How were adverse events evaluated?
Please present the statistical test used to evaluate the normal distribution of the data with its respective p-value.
Considering the findings that were mentioned in the abstract and some that are presented in the results, the statistical analysis should be more robust. A multivariate regression analysis should be run.
Results
Lines 166-173. This information is not consistent with that displayed in Figure 1.
It is mandatory to present adverse events in all clinical trials.
Line 184. This information demands a regression analysis.
Lines 191-193. Table 1 compares these variables in the two groups and establishes the statistical differences between them. To establish associations between these variables and control for confounding variables, a more complex analysis is required.
Line 203. Correlated? In the statistical analysis, the performance of a correlation analysis between the variables is not indicated.
Lines 212-216. The analysis performed to obtain these results is not described.
Discussion
Lines 220-221. Please present the references.
Lines 221-222. Did the same happen with the articaine group?
Line 241. 'Other studies' is described but only one study is referenced.
Line 244 and line 250. Previous observations should be considered.
Line 251. This information was not presented in the results.
Lines 276-278. The implications of this attrition on the results should be mentioned.
Line 285. Present the recent studies that are mentioned.
This work has limitations that were not described. Some of them were mentioned above.
Conclusions
They should focus objectively on the findings.
It is recommended to review the grammar and language.
Author Response
Thank you for your reviewing.
Please see the attachment for our detailed answers to your comments.
Reviewer 3 Report
This study is interesting as it compares two pulp dressing materials in terms of pain management in acute pulpitis cases. however, it would be better to include a control group of pulpotomy cases to achieve more accurate conclusions.
In line 121: Better to use "before treatment" or similar terms instead of “On day0” to make it clearer to the readers.
Was rubber dam used during the procedure?all the details need to be mentioned.
Author Response
Point 1: "it would be better to include a control group of pulpotomy cases to achieve more accurate conclusions."
Response 1: We are fully aware of this limitation and acknowledged it in the discussion (line 269-277). We also explained the reasons behind that decision: national recommandations do not permit the lack of a medication. We could have tried to include of a group treated by pulpotomy without medication but this would have been another protocol which would have been considered as experimental, as opposed to our protocol which evaluated already existing and validated treatments. This could have led to a rejection by the ethics committee.
Including another group is not possible at this stage of the research.
We hope you'll understand the reasons behind this protocol and won't reject it for its limitations despite the obtained results that are still of scientific interest.
Point 2: "In line 121: Better to use "before treatment" or similar terms instead of “On day0” to make it clearer to the readers."
We added "before treatment" to make it clearer to the readers.
Point 3: "Was rubber dam used during the procedure?all the details need to be mentioned."
Response 3: We used rubber dam when it was possible. However, many teeth were so structurally compromised that using a rubber dam was not possible or would not permit to achieve isolation from saliva. In those cases, cotton rolls and aspirators were used to achieve the best isolation possible. Since all patients were included in an emergency dental care unit, we had no time to realize a pre-endodontic restoration to fix a rubber dam. The full endodontic treatment (which is not part of this study but was proposed to all patients) was always performed using a rubber dam, after pre-endodontic restoration if necessary.
We added the following sentences in the protocol (line 133):
"-The tooth was isolated from saliva, using rubber dam when the use of a clamp was possible. If using a rubber dam was impossible (e.g., structurally compromised tooth) and since pre-endodontic restoration was not manageable in emergency, isolation was obtained using cotton rolls and dental aspirator."
We thank you for your constructive suggestions.
All the best.
Reviewer 4 Report
This a very interesting paper since it provides scientifically sound similarities/differences in effect of these two very popular drugs. Therefore, daily clinicians will have a supplementary criterion when selecting a specific drug.
Author Response
Thank you for your interest in our paper.
All the best.
Reviewer 5 Report
Dear Authors, I commend your research and this paper, and suggest several minor corrections to the text. I also recommend that your paper after these corrections be considered for publication. All the best

Author Response
We thank you for the minor corrections. We applied all of them and feel the changes made the manuscript better.
Thank you for your help.
All the best.
Round 2
Reviewer 1 Report
The impovements in the manuscript respond many reviewers points, but I keep asking the author, why did not use a biological dressing material as dressing among treatment sessions? The cost of these materials could be a problem?
The pacient pain was evaluated 1, 3 and 7 days, why did not made the pulpectomy treatment in day 1?
Author Response
Point 1: The impovements in the manuscript respond many reviewers points, but I keep asking the author, why did not use a biological dressing material as dressing among treatment sessions? The cost of these materials could be a problem?
Answer 1: Yes, our aim was to test "basic" materials. Anesthesics are always available and could even be obtained from the syringe used for anesthesia while eugenol is likely present in every dental office as a liquid and thus both are easy to use on a cotton pellet. MTA and biodentine are likely efficient dressings too considering their anti-inflammatory properties and the litterature confirms it, but they present some limitations as emergency dressings: their cost as you stated, their slightly more complex protocol (especially biodentine), the need to remove it after since they are used as cements and not droplets on cotton and finally the fact that they may not be available in all dental offices.
We clarified our decision to compare eugenol and articaine by adding "two inexpensive, easy to use and " to the following sentence:
"With the disappearance of some medications, we wanted to validate and compare the efficiency of two inexpensive, easy to use and easily available products used for analgesia which are present in every dental office: articaine and eugenol. "
Point 2: The pacient pain was evaluated 1, 3 and 7 days, why did not made the pulpectomy treatment in day 1?
Answer 2: To make it short, we could not receive the patients for the pulpectomy in such a short delay due to organizationnal constraints.
As stated in the manuscript, patients were included in an emergency dental care unit. This means patients were able to receive the emergency treatment (pulpotomy) quickly without an appointment.
However, the follow-up (pulpectomy and full endodontic treatment) was performed in a classical dental car unit, in which the appointment is not immediate and sometimes takes place a few weeks later depending on the amount of patients already waiting; up to a month as stated in our "Method" paragraph.
This is not a problem as long as the emergency treatment is efficient enough to relieve pain. If it is really needed, the pulpectomy can be performed in emergency: this is what happened to the two excluded patients from the articaine group as stated in the beginning of the "Results" part of our article.
We hope this answers your last questions.
Reviewer 2 Report
The authors adequately resolved the observations made; therefore, the publication of this manuscript is recommended.
Author Response
Thank you for your support.
Round 3
Reviewer 1 Report
The manuscript was improved, but still laking discussion about the main pourpous of the study, once many materials and data in the literature demonstrates better clinical aproaches for pulpotomy treatment.
Author Response
Point 1: "The manuscript was improved, but still laking discussion about the main pourpous of the study, once many materials and data in the literature demonstrates better clinical aproaches for pulpotomy treatment.
Answer 1:
We added the following sentence (in italic) to the discussion, echoing the similar changes we already made to the introduction following previous rounds of reviewing:
"As stated in the introduction, we had to comply with national recommendations and were not able to use an inactive product as control. We choose to evaluate two products that are cheap, widely available, and traditionally used for short-term pain management. We did not evaluate the most recent but experimental procedures such as long term pulpotomy using MTA or Biodentine whose goal is mostly long-term survival which was not the purpose of this study. However, the higher efficiency of eugenol compared to articaine, and the overall excellent efficiency of the procedures are still important and valid results. "